# Infant-directed input and literacy effects on phonological processing: Non-word repetition scores among the Tsimane'

**Alejandrina Cristia**[1]*, **Gianmatteo Farabolini**[2], **Camila Scaff**[3], **Naomi Havron**[1], **Jonathan Stieglitz**[4]

**1** Laboratoire de Sciences Cognitives et de Psycholinguistique, Département d'Etudes cognitives, ENS, EHESS, CNRS, PSL University, Paris, France, **2** Department of Experimental and Clinical Medicine, Università Politecnica delle Marche, Ancona, Italy, **3** Institute of Evolutionary Medicine, University of Zurich, Zurich, Switzerland, **4** Institute for Advanced Study in Toulouse, Université Toulouse 1 Capitole, Toulouse, France

\* alecristia@gmail.com

**Data Availability Statement:** Data are held in public repository: https://osf.io/bhg94.

**Funding:** AC acknowledges financial and institutional support from Agence Nationale de la

## Abstract

Language input in childhood and literacy (and/or schooling) have been described as two key experiences impacting phonological processing. In this study, we assess phonological processing via a non-word repetition (NWR) group game, in adults and children living in two villages of an ethnic group where infants are rarely spoken to, and where literacy is variable. We found lower NWR scores than in previous work for both children (N = 17; aged 1-12 years) and adults (N = 13; aged 18-60 years), which is consistent with the hypothesis that there would be long-term effects on phonological processing of experiencing low levels of directed input in infancy. Additionally, we found some evidence that literacy and/or schooling increases NWR scores, although results should be interpreted with caution given the small sample size. These findings invite further investigations in similar communities, as current results are most compatible with phonological processing being influenced by aspects of language experience that vary greatly between and within populations.

## Introduction

While the ability to use language is universal across human cultures, there is recurrent debate on whether linguistic processing varies across populations, and if so, why. In this article, we present preliminary data bearing on two factors potentially leading to population variation. We used an "imitation game" to elicit repetition of non-words in children and adults in two Tsimane' villages, where linguistic input is scarce and literacy is variable.

The primary motivation for this study was to assess a potential long-term impact of language input on the emergence and strength of certain phonological representations. The Tsimane' have garnered some attention recently because a behavioral observation study suggested that young children, particularly infants, were spoken to very rarely (about 1 minute per hour, [1]; see also [2]), and potentially less than in many other societies where infant

Recherche (ANR-17-CE28-0007 LangAge, ANR-16-DATA-0004 ACLEW, ANR-14-CE30-0003 MechELex, ANR-17-EURE-0017) and the J. S. McDonnell Foundation Understanding Human Cognition Scholar Award. JS acknowledges IAST funding from the French National Research Agency (ANR) under grant ANR-17-EURE-0010 (Investissements d'Avenir program). url ANR: https://anr.fr/ url McDonnell Foundation: https://www.jsmf.org/ The funders had no role in study design, data collection and analysis, decision to publish, or preparation of the manuscript.

**Competing interests:** The authors have declared that no competing interests exist.

language input has been documented quantitatively [3]. Additionally, the present study contributes to a second strand of research, bearing on the impact of literacy (and/or schooling) on language processing. Since access to schooling varies in the Tsimane' territory and instruction in Bolivian schools is mostly in Spanish, whereas children tend to be raised monolingual Tsimane', the adult prevalence of literacy is about 18% [4]. We review previous work and motivate the present study on each of these research lines in turn.

## Early input effects on phonological processing

The first line of research our data bear on is the impact of language input on the emergence and strength of certain phonological representations. A hypothesis holds that, when learning to speak or sign, as well as to perceive speech/sign, humans likely develop short-hand-like phonological representations that make that process very efficient. This general hypothesis is shared among many classes of theories: those in which human language acquisition crucially relies on domain-specific processes unique to language [5], as well as theories where this process simply re-utilizes neural networks that are shared with other species and/or other cognitive processes [6]. Despite potentially different ways of articulating the process (e.g., "selection of language-specific symbolic representations" versus "neural entrenchment"), all assume that something happens in the mind/brain of infants whereby processing of native categories and sequences is rendered more efficient through experience. Also, all theories of phonological development assume that input plays a pivotal role in acquisition: Since the precise phonological inventory varies across languages, infants cannot be pre-programmed with specific phonological categories. Instead, each child must learn the phonological inventory of her language(s) from her language-specific experiences (including those in utero).

Although all theorists agree that input must play at least this role, the extent to which the actual learning or acquisition process is proposed to rely on the input varies widely (see [7] for discussion). For instance, some theorists propose that a caregiver's responses to an infant's babbling are crucial to phonological development [8], in which case the input that is directed to the child must affect phonological development to a greater extent than input that is merely overheard.

In addition, theories can be classified in terms of whether phonological development is driven by lexical development or independent from it. For the former, input must also affect phonology indirectly, as we explain next. Lexically-driven theories come in many flavors. Some have proposed that infants extract short-hand-like phonological representations from a few words they know well [9]; others that children rely on minimal pairs (words differing on a single sound, e.g. since "pin" and "tin" mean different things, /p/ and /t/ are different sounds in English; [10]); yet others that those phonological representations emerge as the children's vocabulary becomes dense enough [11]. Regardless of the precise process whereby phonological categories are affected by lexical development, it follows that, according to any of these theories, any factor that impacts lexical development should then indirectly affect phonological development. One such key factor is children's directed input. There is now a great deal of evidence that individual and sociocultural variation in lexical development is at least partially explained by the quantity and quality of the interaction the child is involved in [12–15]. Thus, if phonological development is driven by lexical development, and lexical development is driven by infant- and child-directed input, then environments that are associated with little infant- and child-directed input should also have less robust or later-developing short-hand-like phonological representations.

One measure of children's and adults' short-hand-like phonological representations is non-word repetition (NWR), a task in which the participant hears phonologically grammatical but

meaningless words (e.g., *beng* in English) and they must repeat out loud each as similarly to the model as possible. These studies often employ 12-30 items varying in length from 1 to 5 syllables (most commonly 2-4 syllables). NWR scores are typically estimated as the percentage of items repeated correctly (as judged by the researchers). NWR scores have been found to correlate with other measures of phonological processing concurrently [16, 17] and to be a robust predictor of later literacy [18, 19]. Additionally, and in line with theories proposing that phonological processes depend on lexical development, NWR correlates with vocabulary size (e.g., [20]).

In the present study, we report on NWR data collected from both adults and children. If directed input is crucial for the development of short-hand-like phonological representations, then Tsimane' children's NWR scores will be lower than that among children growing up in populations where infant-directed input is more common.

Given extant evidence suggesting that phonological acquisition is governed by a critical period closing around the second year of life [21], it is possible that children do not come to accumulate sufficient directed input in this period, and thus their phonological processing follows a different developmental pathway which may not include the emergence of short-hand-like phonological representations. A similar argument can be made for any critical age threshold. For instance, imagine that the most sensitive period for phonology exposure starts closing at age 12 years. Children who have received little input from birth to 2 years of age can only catch up with children who have received more during the same period if the former receive more linguistic input than the latter during the 2-12 years period. Finally, some theorists believe there is no critical period, but instead a variety of experiences shapes phonological representations throughout the life span (including literacy and/or schooling, as described in more detail in the next section). For instance, some research suggests that non-word repetition scores increase with age before children start learning to read [22–25], as they start to read [26], as well as afterwards [27, 28]. These different views make diverse predictions regarding the performance one expects to find in a population where infant-directed speech is rare: If there is a critical period in infancy, then scores should be much lower; if this critical period ends later, then we may observe a small difference; and if there is no closure, we can expect to see NWR scores matching those found in other populations at some relatively late age.

To test these predictions, we would need to know what are the levels of NWR scores expected at each age in populations where infant-directed speech is more prevalent. Unfortunately, no meta-analytic review is available on the NWR literature, and performing one was beyond the scope of the present paper. We combined our previous knowledge of the literature and systematic searching to yield a sample of 17 studies that we could interrogate further [22–25, 27–39].

Ten of them, represented in Fig 1, provided word-level NWR scores among monolingual, typically-developing children aged 4 to 12 years [22–25, 27, 28, 31–34]. This figure reveals a wide range of scores across studies, with the average percentage of non-words that are correctly repeated varying between 25 and 95%. Additionally, the divergent slopes visible in Fig 1 suggest that age effects varied across studies. For instance, children tested by [33] increased 9% between 4.5 and 7.5 years, whereas children in [25] jumped by 20% over roughly in the same period. Cross-study variation may relate to differences in the complexity of the non-words employed, since shorter non-words should lead to higher NWR scores than longer ones. That said, when studies report NWR scores separately for items varying in length, scores for mono- and bisyllabic items can be a great deal higher than that for longer items [31] or only slightly so ([25]; see Fig 4).

We interrogated this literature further to check for the possibility that effects of infant-directed input quantities may already have been reported on. Input variability has not been

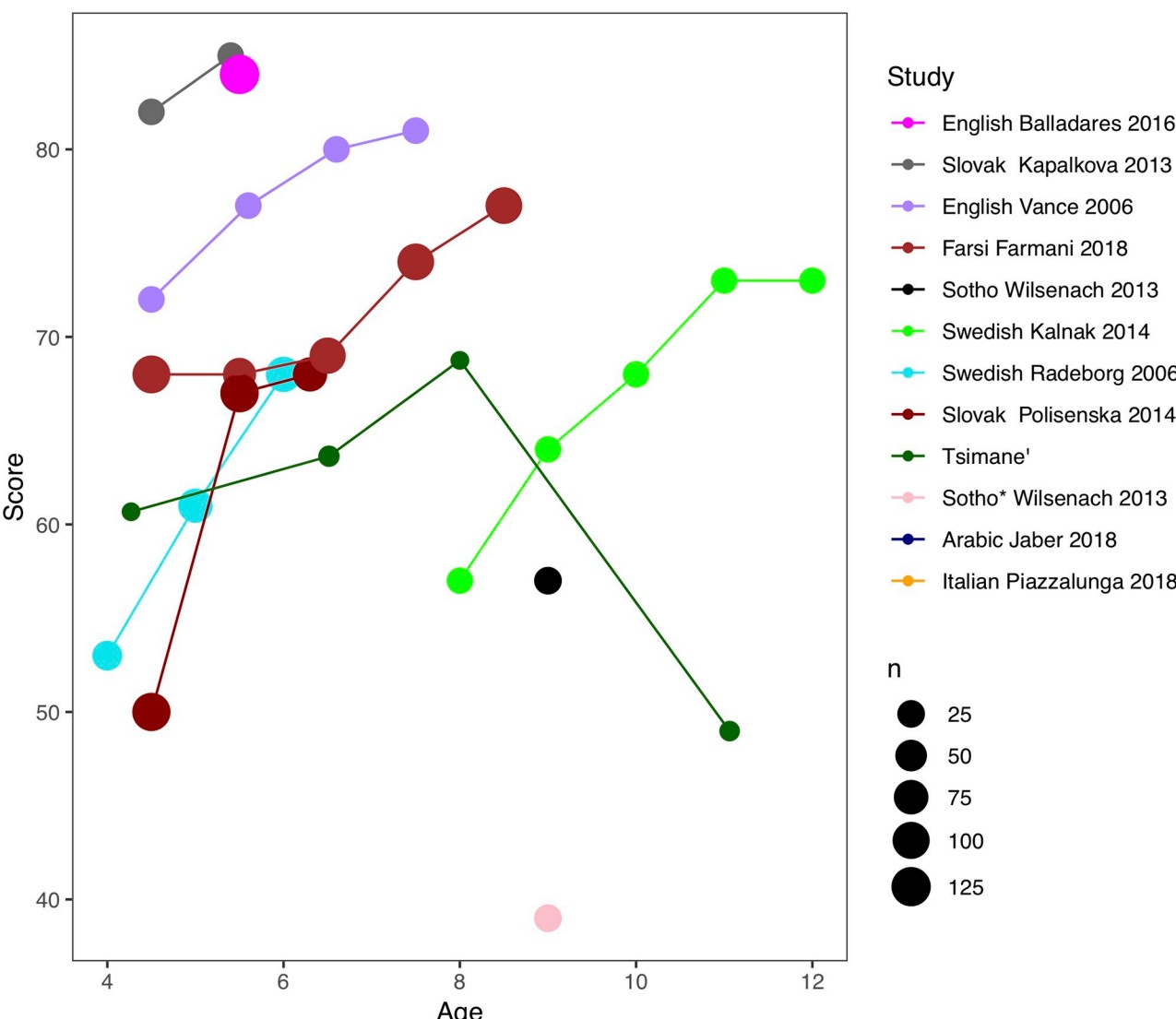

**Fig 1. NWR scores as a function of age (in years), study (first author and year regardless of the number of authors).** Study legend is sorted from highest to lowest average to facilitate linking. The size of the circle indicates sample size.

studied, to our knowledge, but two potential proxies of input have: Monolingual status, and socio-economic status. We set aside the literature comparing monolinguals against non-monolinguals because this contrast may not only show effects of input differences, but also interference across the languages being learned. Socio-economic status has repeatedly been reported to correlate with differences in quantity of speech directed to children [12, 40]. If any-thing, one could argue that effects of socio-economic status may overestimate input effects since socio-economic status is likely to be correlated with factors that affect performance in any task, such as stress [41]. And yet, our small-scale systematic review revealed that correla-tions between socio-economic status and NWR scores tend to be too weak to be detected with the sample sizes used in previous work. To be precise, seven of the 17 papers in our review did not discuss SES differences at all [22, 24, 29, 30, 33–35]. Among the remaining 10, 5 did not analyze potential NWR scores differences as a function of SES, often citing work suggesting

that NWR scores did *not* vary as a function of SES; [31, 36–38, 42], and the remaining 5 did analyze their data and reported no significant effect [23, 27, 28, 39, 43].

To sum up, one key motivation for this study was to investigate NWR scores in a population in which infant-directed speech has been previously found to be rare. If input during infancy plays a key role for the development of short-hand-like phonological representations (directly and/or indirectly via lexical skills), then we should observe low levels of NWR scores in Tsimane' children (and perhaps also Tsimane' adults, if there is a critical period or similar levels of language exposure are never achieved; see also the next section). However, it is also possible that directed input does not play a key role in the emergence of short-hand-like phonological representations directly, and its effect on phonology via the lexicon is too small, in which case the levels of NWR scores observed here may be comparable to those found in previous work.

## Literacy and/or schooling effects on phonological processing

We now turn to the second strand of research to which this study contributes, regarding the impact of literacy on language processing. As mentioned previously, some researchers posit that experiences even beyond two years of age can have a profound impact on language representations and skills. One type of experience that has been connected with phonological processing specifically is literacy. Illiterate adults perform a range of linguistic tasks less well than literate adults, including phoneme deletion (e.g., removing the first sound in "tin" yields "in"; [44, 45]), and phonological fluency (e.g., say as many words beginning with "p" as you can; [46–48]).

Before going further, it may be relevant to discuss to what extent these effects are due to literacy per se, rather than schooling more generally. Notice that literate and illiterate adults typically do not only differ in their ability to read and write, but also in the fact that the former have been more extensively schooled, and thus trained in performing arbitrary tasks and tests, as discussed in [49]. [49] included four groups of participants: readers, nonreaders, non-literates who attended school, and self-taught readers. The authors tested the effect of literacy versus schooling on working memory, and found an effect of literacy per se on working memory. However, this study did not test phonological processing specifically. A separate but related issue relates to the open question of whether literacy only affects performance on tasks that require conscious manipulation of speech sounds and units (such as phoneme deletion and phonological fluency), or whether it also affects more implicit phonological processing involved in everyday language processing and learning, as discussed, for instance, by [50] and [51]. We return to this when discussing [52] below.

We have only found six studies on NWR as a function of literacy and/or schooling among adults [47, 48, 52–55], summarized in Table 1, which also includes some results of studies reporting NWR scores among adults collected for other purposes (to study phonological processing of grammatical versus ungrammatical non-words; [56]). Table 1 adopts the terminology used in each of the papers to describe the populations they sampled from. [47] classified as illiterate only participants who met several conditions including not reading any kind of written material (except for their names in some cases) and being born of illiterate parents; in contrast, literates had acceptable performance on tests of reading and writing. [55] tests a subset of participants of [47] and further adds that functional illiterates (with exposure to reading and writing, likely through schooling) were excluded. In [54], non-readers and readers both self-identified as illiterates; among these, non-readers were participants who did not read at all, whereas readers could read at least one word. Controls, in contrast, attended at least 5 years of school. Illiterates in [53] had never attended school and although they could sign their initials,

**Table 1. Summary of comparisons in non-word repetition (NWR) between adults varying in their level of literacy and/or schooling.** Studies are designated by the last name of the first author (dSL = de Santos Loureiro) and the publication year regardless of how many authors there are, with rows sorted by the paper's year of publication. Language indicates the participants' native language(s); Br. indicates data collected in Brasil rather than Portugal. In Group, we use the terminology of each paper to identify each of the tested group (see footnote 1 for details). n indicates the number of included participants. Age indicates average age (SD) [age range], in years. NWR score (in percent) indicates the overall score collapsing across all non-words, except for Kolinsky 2018 and dSL 2004, where we averaged NWR scores across non-word length means. dSL 2004's controls were only tested with long non-words.

| Study | Language | Group | n | Age | Score |
|---|---|---|---|---|---|
| Reis 1997 | Portuguese | illiterate | 20 | 61 | 77 |
| Reis 1997 | Portuguese | literate | 10 | 58 | 98 |
| Castro-Caldas 1998 | Portuguese | illiterate | 6 | 65 (5) | 33 |
| Castro-Caldas 1998 | Portuguese | literate | 6 | 63 (6) | 83 |
| dSL 2004 | Portuguese (Br) | non-readers | 68 | 43 (11.8) | 72 |
| dSL 2004 | Portuguese (Br) | readers | 29 | 43 (11.3) | 78 |
| dSL 2004 | Portuguese (Br) | control* | 50 | 31 (7.8) | 82 |
| Kosmidis 2006 | Greek | 0 years | 19 | 72 [63-92] | 80 |
| Kosmidis 2006 | Greek | 1-9 years | 20 | 70 [56-85] | 96 |
| Kosmidis 2006 | Greek | 10-16 years | 15 | 62 [55-74] | 99 |
| Tsegaye 2011 | Amharic | illiterate | 11 | [25-45] | 44 |
| Gallagher 2014 | Quechua-Spanish | literate | 21 | [20-40] | 99 |
| Kolinsky 2018 | Portuguese | before training | 7 | 40 [22-64] | 64 |
| Kolinsky 2018 | Portuguese | after training | 7 | 40 [22-64] | 74 |
| This study | Tsimane' | non-readers | 7 | 30 [15-47] | 54 |
| This study | Tsimane' | readers | 6 | 36 [20-60] | 67 |

they failed in a reading test consisting of identifying graphemes and reading a paragraph. Literates in this study not only had attended school (1-9 years) but also reported reading regularly. Illiterate participants in [48] could not read or write in a test including letters and simple words, whereas literate participants had 7-10 years of schooling. Finally, [52]'s study will be introduced below as having a pre-post design on the same participants, who before training had attended school irregularly 0-2 years. To simplify discussion, we collapse across illiterates, non-readers, zero years of schooling, and "before literacy training", but interested readers can inspect Table 1 for raw data on these separate categories. As in the childhood literature reviewed above (see Fig 1), studies on adults report widely variable levels of NWR scores.

Among these studies, [52] is particularly relevant in terms of decorrelating literacy and schooling. [52] reports on a longitudinal study in which 8 previously illiterate adults received 14 weeks of reading training, and six of them learned how to read in this process. NWR data was collected several times through this period, each time having participants repeat non-words with increasing syllable complexity (CVs, then CCVs) and length (1-6 syllables). NWR data was available for 7 participants, whose average score on a subset of 8 items increased from 21% to 38%—however, this was for a subset that the authors thought had no ceiling effects. Indeed, it was found that participants could repeat CV monosyllables with 100% scores; in contrast, 5- and 6-syllable long items with CCV yielded 0% scores in both the initial and final sessions. Results for the monosyllables suggest that NWR per se is achievable by adult non-readers. Although scores increased over the course of the training, the authors found that NWR scores did not correlate with reading skills at the individual level. The authors came to the conclusion that reading may improve phoneme awareness, which in its stead can be co-opted as an attentional strategy during NWR. In addition, the training (3 weekly classes of 2h each over 14 weeks) and tests (3 tasks on reading, 3 tasks on meta-phonological skills, and the NWR, administered 5 times over a 17-week period) were rather demanding, and thus it is

conceivable that participants' test-taking skills may also have improved over the course of the training. Thus, even in this study with a longitudinal, pre-post design, we do not find unequivocal evidence that literacy training leads to the development of short-hand-like phonological representations, since higher NWR scores may have been due to indirect attentional effects (as suggested by that study's authors) or general task effects (given the intensive practice).

There is one more issue affecting the interpretation of most previous work. In nearly all previous studies, participants repeated both words and non-words mixed together (with the exception of [55], who blocked words and non-words in separate trials; and [52] who presented only non-words in this task). This may have caused confusion as participants concurrently perform a lexical decision task, which is meta-linguistic, and which may have interfered with the actual repetition. As a result, it is possible that the exact same activity presented as a game containing only made-up words more accurately measures phonological processing among illiterate adults, leading to higher NWR scores among illiterates than expected given previous findings.

## The current study

In a nutshell, we tested children and adults in a society where infants are rarely spoken to, and where literacy is variable. As a result, it is possible that phonological processing is different due to the low levels of directed input in infancy, as well as because literacy (and any skill associated with it) has not served to reinforce and/or maintain short-hand-like phonological representations among our illiterate participants. As discussed in subsequent sections, the sample size of the present study is fairly small and analyses were not pre-registered, and should be considered exploratory. We inspect our data in three main ways.

First, by comparing the NWR scores of Tsimane' participants against those of participants in previous literature, we aim to shed light on the potential long-term effect of low levels of infant-directed input on phonological processing. Second, by comparing the NWR scores of Tsimane' adults against previous similar work, we aim to measure joint effects of low levels of infant-directed speech and variable literacy on phonological processing. Finally, we compare scores across subsets of our participants to examine phonological processes underlying non-word repetition: We compare children and adults to look at developmental changes; self-reported readers and non-readers to assess the potential impact of literacy; and participants having completed higher versus lower grades to study effects of formal education.

Considering all of these effects together is important in the general move away from statistical significance as the main criterion for judging noteworthiness, and towards contextualized reading of the size of effects (e.g., [57]). By studying multiple factors together, we can discuss their relative importance. For instance, proponents of a critical period for phonology that closes at a certain age may argue that any age-related changes after this age are marginal compared to the key knowledge that is acquired before then. Similarly, proponents of literacy-driven changes in phonology may concede that schooling affects general performance, but argue that this effect is smaller than that of literacy per se. Instead of focusing on a subset of effects, we present all relevant ones, and further share our data to enable reuse.

## Tsimane' language and community

The Tsimane' are an indigenous group residing in the forest, riverine, and savanna areas of lowland Bolivia, in the Beni department (for a map, see Fig 1 of [58]). While they are experiencing a fast market integration into broader Bolivian society (increasing town visits, schooling, knowledge of Spanish, wage earnings and consumption of market goods; [59]), almost all of the food the Tsimane' consume comes from horticulture, fishing, and hunting.

| Consonants | Labial | Dental/Alveolar | Post-alveolar | Palatal | Velar | Glottal |
|---|---|---|---|---|---|---|
| Aspirated | pʰ (345) | | tsʰ (114) | tʃʰ (1888) | kʰ (1260) | |
| Plain | p (1403) | ʈ (59), t (4041) | | tʲ (1666) | k (8380) | ʔ* (11213) |
| Voiced | b (2393) | ɖ (4), d (850) | | dʲ (3051) | (g 62) | |
| Affricate | | ts (832) | tʃ (681) | | | |
| Fricative | f (697) | s (3225) | ʃ (1277) | | x (8411) | |
| Nasal | m (3322) | ɳ (41), n (4022) | | ɲ (851) | | |
| Appr/Lat | ʋ (2489) | l (152) | | j (5421) | | |
| Tap | | ɾ (2636) | | | | |

| Vowels | Front | Center | Back |
|---|---|---|---|
| High | i (12617), ĩ (255) | ɨ (4637), ɨ̃ (309) | |
| Mid | e (9200), ẽ (153) | ə (2354), ə̃ (33) | o (5066), õ (397) |
| Low | | a (13163), ã (489) | |

**Fig 2. Phonemic inventory in Tsimane' obtained by crossing several sources, and the number of tokens of that type found in Gill (1999)'s dictionary.** /g/ occurs in loanwords and is not considered native. The glottal stop may actually be a feature of the preceding sonorant, and not a phoneme (it only occurs after vowels and nasal stops, never in syllable-initial position; no stop visible in spectrograms even when the word ending in this sound was followed by a vowel-initial word). Appr/Lat stands for Approximate or Lateral.

There are about 16,000 individuals (about half of them being under 15 years of age) living in about 90 villages [59]. Brief summaries of various aspects of Tsimane' history and culture can be read in [60].

Data were collected in two Tsimane' villages. Both villages had a school, and there was school every weekday when it did not rain while we visited. However, most Tsimane' adults sampled here reported relatively low levels of completion of formal education and low levels of literacy for themselves and their children (see Participants section for details on the included sample, and [4] for more data on literacy in the population at large).

Most Tsimane' are monolingual in the Tsimane' language, which is unrelated to other South American languages (with the exception of the closely related Mosetén; [61]). The language has been spelled as Tsimane', Tsimané, Chimani, Chimané, Chimane.

There are no publications on Tsimane' phonology specifically. However, there is important information in a dictionary and a grammar by the missionary Wayne Gill [62, 63], a footnote in [64], as well as a variety of materials deposited by Sandy Ritchie in the Endangered Languages Archive at SOAS University of London [65]. We unfortunately did not have access to a phonemic statement mentioned in [63], but it seems likely that there is a vowel length distinction that is not coded in the writing. We crossed all available information to develop a full correspondance table between graphemes and phonemes, which resulted in the phonemic inventory shown in Fig 2, and which allowed us to study phonological statistics from the dictionary and other sources (including [65] and other material found online).

Statistically speaking, Tsimane' syllables are mostly consonant-vowel, with only a few consonants appearing in coda position (including nasal stops, all voiceless fricatives, and the glottal stop—but note this may actually not be a stop), and no onset clusters. An analysis of entries in the Tsimane'-to-English portion of [62]'s dictionary revealed the most common word template was SVF (106 types; S stands for stop, V for vowel, F for fricative); the second most common SVSV (95 types); and the third SVSVF (84 types). Other word forms sampled from the rest of the distribution: AVS (where A stands for approximant; 32 types); FVFVS (18 types); SVSVSVSV (7 types); all other word forms had a prevalence lower than 5 types in the dictionary.

There seem to be no monomorphemic words longer than 3 syllables in Tsimane'. The only 4-syllable and 5-syllable words in the dictionary are verbs containing the infinitive morpheme /aki/ (e.g., "carijtaqui" *to work*), and thus the root was only 2-3 syllables long. Nonetheless, we also considered 4-syllable items among the stimuli for completeness and comparability with other studies.

## Methods

This paper was written using the rticles package [66] and RMarkDown [67] in R [68] running on Rstudio [69]. It can be downloaded and reproduced using the data also available from the Open Science Framework, https://osf.io/bhg94.

We had created a pre-registration for this study prior to departing for Bolivia (see Supplementary document https://osf.io/b3sfa/, section 1, for more information on the field work setting). However, our methods deviated considerably from the pre-registered ones starting with the procedure: It soon became obvious that eliciting non-word repetitions by playing back sounds from a computer, one person at a time, was not feasible, and would yield absolutely no data from children. This lead us to test people in groups, which resulted in data structured differently from the data originally expected. We note here only what actually was used in the end. All analyses must thus be considered as exploratory (meaning that *p*-values cannot be interpreted as based on hypothesis testing), and the analyses as a whole should not be considered pre-registered.

### Stimuli

See Supplementary document https://osf.io/b3sfa/ (Section 1) for a detailed historical recount of stimuli creation. The first list of nonwords to be used in the field was generated through discussion with a native Tsimane'-speaking research assistant based on a prior list. Any words recognized as real by the Tsimane'-speaking research assistant were excluded. One nonword that was used for the first groups was removed after it became obvious that it was sometimes interpreted as a real word in Spanish—"dos" (the number two).

The actual stimuli list continued to evolve over the course of fieldwork, in order to add stimuli of greater complexity, and as systematic mispronunciations emerged that could be attributed to a cause other than the repetition game. Specifically, initially several items had sounds where phonetic implementation in Tsimane' diverged from AC's native language (Spanish), namely /u,w,f/. Although AC is phonetically trained and tried to pronounce these with the Tsimane' implementation, it was uncertain whether NWR errors reflected pronunciation failure on her part or repetition difficulties. Therefore, these items were progressively replaced by generating new nonwords starting from real words that were extracted from the grammar [63]; or the dictionary [62].

As a result, the precise list of items used changed throughout the study. These are the items that were presented to each group of participants (in pseudo-IPA: /T/ stands for the palatal voiceless affricate; /N/ for the palatal nasal stop; /S/ for the palatal voiceless fricative; /w/ for the labiodental approximant):

- 1: *dos, sin, taf, wik, tadi, bike, kito, boxtum, fisek, potex, wodix*

- 2: *dos, sin, taf, wik, dadi, bike, kito, boxtum, fisek, potex, wodix*

- 3: *sin, taf, wik, dadi, bike, kito, boxtum, fisek, potex, notoxNe, Tedikoti, tipijerax, xaferate*

- 4: *sin, taf, wik, dadi, bike, kito, boxtim, fisek, potex, wodix, notoxNe, Tedikoti, tipijerax, xaferate*

- 5: *sin, taf, wik, dadi, bike, kito, boxtim, fisek, potex, wodix, notoxNe, Tedikoti, tipijerax, xaferate*

- 6: *sin, taf, wik, dadi, bike, kito, boxtim, fisek, potex, wodix, notoxNe, Tedikoti, tipijerax, xaferate*

- 7: *sin, taf, wik, dadi, bike, kito, boxtim, fisek, potex, wodix, ajaSa, oSiso, peside, deSpote, koxtika, notoxNe, Tedikoti, tipijerax, xaferate*

We calculated word shape (defined as sequences of consonants and vowels) and phoneme frequencies using the 16,518 Tsimane' lexical entries in the Tsimane'-to-English portion of [62]'s dictionary. In terms of overall shapes, we found 614 unique types in the dictionary (with 351 that were hapaxes, having a frequency of 1). The frequencies (number of tokens for that type divided by number of tokens of all types) and ranking of the shapes (from most to least frequent) present in our stimuli were as follows. CVC ranked sixth with 3.99% of tokens having this type (in each case, we give examples of non-words in our stimuli having this shape: *sin, taf, wik*). CVCV ranked ninth with 3.04% (e.g., *dadi, bike, kito*). CVCCVC ranked first with 8.19% (e.g., *boxtim*). CVCVC ranked second with 9.24% (e.g., *fisek, potex, wodix*). VCVCV ranked 47th with 0.36% (e.g., *ajaSa, oSiso*). CVCVCV ranked eighth with 3.11% (e.g., *kijeki*). CVCCVCV ranked 10th with 2.96% (e.g., *deSpote*). CVCVCCV ranked 13th with 1.84% (e.g., *koxtika*). CVCVCVCV ranked 20th with 0.92% (e.g., *Tedikoti*). CVCVCVCVC ranked 16th with 1.57% (e.g., *tipijerax*). Thus, our stimuli sampled a wide range of frequencies, from the two most frequent word shapes (CVCCVC, CVCVC) to a shape ranked 47th (VCVCV) out of a potential 351 non-hapax shapes.

As for phonemes, we found 118,627 tokens of 41 phoneme types in the inventory (see Fig 2). We reasoned that the frequency and ranking of individual phonemes was less informative than the average phoneme frequency for each one of our non-words (i.e., for *sin*, the average of the frequency of /s/, /i/ and /n/). Our items' average phoneme frequency ranged from 2.57% to 7.95%, with a mean at 5.6%.

For descriptive analyses, we classified items as a function of two structural properties (length in number of syllables and whether they contained any closed syllables), as well as two frequency properties (shape frequency and average phoneme frequency).

## Procedure

The activity was presented as a group game, which started when AC said an item and each member of the group repeated that item when AC held a recording device (Olympus WS811) closer to that group member. There was always at least one Tsimane' research assistant in a given group, and she was always asked to repeat the nonword first in the first trial. This served as a model of a natively-pronounced repetition. The research assistant sometimes made mistakes in the repetition, in which case AC said the item again until the research assistant's repetition was accurate. When another participant incorrectly repeated the word, AC said the nonword again but did not necessarily ask the same person to repeat. When doing the offline coding, however, we noticed that many cases of incorrect repetitions by participants were not followed by AC repeating the item (i.e., were not detected as incorrect repetitions on the fly).

Nonwords were typically presented in order of increasing length, which is also in increasing complexity according to previous work. Later items may benefit from practice effects, but scores may also decline over time due to fatigue or interference of previous items in memory.

All NWR tasks in the literature are administered one participant at a time. One downside of presenting items in a group as we did is that when someone misspoke it, this could serve as the

model for another person, particularly since AC was clearly not a Tsimane' speaker. This will be taken into account in preliminary analyses (see Preprocessing section).

## Participants

Institutional IRB approval was granted by University of New Mexico (HRRC # 17–262), as was informed consent at three levels: (1) Tsimane' government that oversees research projects (Gran Consejo Tsimane'), (2) village leadership and (3) study participants. We made a public presentation at each village where we explained the general goal of the research, to study language acquisition, and demonstrated some of the methods. After this, people visited our camp and/or we visited them in the context of other studies. At this time, they were asked whether they wanted to additionally participate in this study, which took between 1.5 and 6.5 minutes. In addition, participants' reactions were monitored to ensure they participated in their own terms (e.g., participants could stop repeating the items). Participants were not compensated for participating in the NWR game specifically, and thus there was no gift to promote higher NWR scores. Instead, participants were compensated in lump sum for the battery of protocols they participated in. Participants consented verbally, for themselves and for their children. This verbal consent/assent procedure was approved by the above-mentioned IRB.

Groups were typically formed by members of one family (mother and one or more children), and sometimes members of two families, in addition to one or both Tsimane' research assistants and another investigator (CS), and sometimes additional children who followed us on visits or visited our camp.

Data were collected from seven groups of between three and nine participants, of which at least one participant was always one of the Tsimane' research assistants. Although children as young as one year of age were present, they seldom attempted to repeat the items. The youngest children in a given group almost always refused to talk, or if they did repeat some of the shorter items, they eventually stopped as items got longer, and thus most data come from older children and female adult family members. We have data from a total of 16 children (mean age 7.07, range 1-12 years, 11 female) and 13 adults (mean age 32.46, range 15-60 years, 11 female). We obtained age estimates by asking participants their date of birth and crossing the information provided with the census information that the Tsimane' Health and Life History Project has been updating for over a decade [59].

We also asked what the highest grade of schooling completed was. Tsimane' children typically begin first grade at 5 years of age. Children can advance one grade each year, except if they repeat a grade. We did not ask how many years they attended school, but rather what was the highest grade completed. The highest grade completed averaged 1.12 and ranged between 0 and 3 among children. Among adults, highest grade completed averaged 2.31 and ranged between 0 and 5.

Finally, we asked whether the participant knew how to read and write on a 3-point scale: not at all (71% of children and 54% adults), a little (29% of children and 23% adults), and yes (no children and 23% adults).

## Scoring

Scoring was done in three phases. First, one or both of the Tsimane' research assistants listened through the file using the Olympus recorder play-back functions, and wrote down for each attempt whether the item was correctly pronounced, and if not what the mispronunciation was. Unfortunately, inspection of this scoring revealed that the actual non-words may have been unclear for the research assistants. For instance, one of the nonwords was *sin*, but the research assistants sometimes wrote zin or zinc (which is the name of a material/element in

Spanish), although the sound /z/ does not exist in Tsimane' (or Spanish). This may be a problem of orthography, since only one of the research assistants was fully literate, and in general in this population orthography is extremely variable. An additional difficulty in coding was that some of the people did not repeat all of the words, and particularly mothers sometimes repeated an item several times to try to get their child to say it. This made attribution of tokens to individuals based on playback from the Olympus recorder difficult and likely errorful. Second, GF (native language Italian) segmented and scored all the items using the audio annotation system Praat [70], independently from the research assistants' coding. Third, AC (native language Spanish) listened through again and resegmented and rescored all items paying close attention not only to the previous two layers of annotation but also to phonetic pronunciation and the phonological and phonetic inventory of Tsimane'. We will refer to the latter scoring as the "judge's" score, since it took into account both layers of previous coding, and will discuss this score only.

NWR was scored both in terms of the whole word (0 if there were any deviations, 1 otherwise) and at the phoneme level. Inspection of previous work does not reveal precisely how the latter is calculated. For this study, we used the proportion of correctly pronounced sounds (not allowing for transposition) divided by the maximum number of sounds (in the model or the participant's production). For instance, if *kito* is pronounced *kirito*, the score would be 4/6; if it were pronounced koti, the score would be 2/4.

Finally, CS listened through all the audio again and verified the individual identity of all the speakers. By again using Praat, it was possible to listen to the same token by different speakers and different tokens by the same speaker in alternation, to be positive about the attributions. As mentioned in the next subsection, since the repetitions were elicited in a group game, the speakers' individual identity could not be verified in 2.37% of cases, which were excluded from analyses.

## Data analyses

**Preprocessing.** There were a total of 523 attempts considered. Out of these, a total of 17 trials were excluded (2 where the audio was too weak to establish the pronunciation; 3 corresponding to documented instances of people not making an attempt; 12 trials in which the speaker's identity could not be determined).

**Justification for using all attempts and not only the first one.** Our reading of previous non-word repetition work suggested that papers vary in whether they allow participants to hear the same non-word several times, and whether all of the attempts are scored or only the first or most similar to the model. Therefore, we checked whether there were differences between first and subsequent attempts (including across multiple days, for our two Tsimane' research assistants and some children who joined the game several times, with different groups). For this analysis, we included only participants who provided multiple datapoints on the same non-words, and inspected only scores on these precise non-words (which is therefore a subset of all the non-words, and does not generalize to the whole set of non-words since phonologically complex words may be repeated more than phonologically simple ones).

A paired t-test between first and other attempts across participants revealed no significant difference, t(11) = 1.12, p = 0.29. Additionally, a logistic mixed model predicting individual attempts' score, declaring as fixed factors age group (child, adult) and whether it was the first or subsequent attempts, with target and participant ID as random factors, did not reveal a significant estimate for attemt (ß = -0.27, SE = 0.28). Since there was no significant difference, and it is unclear that previous work relied exclusively on first attempts, results rely on all the

data (including repeated attempts). For further information, see Section 2 of online Supplementary document https://osf.io/b3sfa/.

**Group size did not affect NWR scores.** Additionally, we assessed whether group size affected NWR scores. A Spearman correlation between NWR scores and group size across all participants was not significant (r(40) = -0.252,p = 0.108). For further information, see Section 3 of online Supplementary document https://osf.io/b3sfa/.

**One person's incorrect repetition did not serve as target for the next person's repetition.** We also checked whether one person's incorrect repetition served as a model for the next person by calculating the proportion of errors where the mistaken production matched exactly the mistaken production of the previous person. This happened in 0.59% of all productions, or 1.53% of productions with errors. Common errors will also lead to mistaken productions looking similar, and yet this percentage is very small, suggesting that even if people imitated each other's errors, this affected a very small proportion of our data.

**Individual NWR scores' reliability is low.** Additionally, we checked for stability in NWR scores by randomly splitting non-words into two within each group, and calculating NWR scores separately for each half. For example, in one run for group 5, the non-words "sin, dadi, boxtim, fisek, notoxNe, Tedikoti, xaferate" may go in one split and the non-words "taf, wik, bike, kito, potex, wodix, tipijerax" would go in the other half. Since by chance one split could end up with all the short non-words and the other split all the long non-words, we repeated this procedure 100 times (runs) to ensure representativity of results. Splitting the non-words reduces the number of attempts considered, and thus some participants did not have data for one of the split halves. When this occurred, these participants were excluded. This led to the loss of 4 participants in 39% of the runs, 5 participants in 56% of the runs, and 6 participants in the remaining 5% of the runs. A Spearman correlation between percentage of items correctly repeated for one half of the non-words versus the other half revealed that NWR scores were not stable for children, because the correlation coefficients were small and the p-value was never below alpha.05 (the p-value was below.1 in only 2% of the runs). For adults, some of the runs suggested some stability in NWR scores across the two halves, with a significant correlation in 14% of the runs, and a marginally significant (alpha.1) correlation in 21% of the runs. Overall, these results suggest that individual measurements are noisy, particularly among children.

**NWR scores vary as a function of phonological complexity.** For descriptive purposes, we show NWR scores as a function of two structural factors, word length and stimulus complexity. Errors are sensible from a phonological viewpoint, as found in a logistic mixed model predicting individual attempts' score from the item's number of syllables, whether or not the item contained a closed syllable, and the participant's age group (child, adult) as fixed effects, and target and participant ID as random effects. Scores were lower for longer than shorter items (ß = -0.87, SE = 0.18) and for items containing at least one closed syllable as opposed to none (ß = -0.78, SE = 0.35). For further information, see Section 4 of online Supplementary document https://osf.io/b3sfa/.

**There is a trend towards NWR scores varying as a function of phonological frequency.** We also explored the predictive value of two measures of frequency, the frequency of word shapes and the average phoneme frequency of each individual item. We fit a binomial mixed model predicting whether an item was correctly repeated or not from the following fixed effects: age group (child, adult), average phoneme and shape frequencies as fixed; item and participant ID were declared as random effects. Although no fixed factor achieved significance, phoneme frequency had a z-value above 1 (average phoneme frequency estimate = 0.27, standard error = 0.25, z = 1.11, p = 0.27), consistent with higher NWR scores for items with higher average phoneme frequency. To provide a purely exploratory illustration, the

proportion of items correctly repeated having average phoneme frequency below the median was 55.61%, whereas for those above the median it was 64.14%.

**Main analyses.** Our first question involves comparing the NWR scores of Tsimane' children against previous developmental literature. As shown on Fig 1, NWR scores in previous work on monolingual children aged between 4 and 10 years is rarely lower than 50% on average.

Answering our second question involves comparing the NWR scores of Tsimane' adults against previous similar work, and looking for variation within this same population. Table 1 summarizes the NWR scores found in previous work comparing illiterates, non-readers, or participants with no formal education against literates, readers, or participants with higher levels of schooling. If NWR scores found among the Tsimane' who self-report being unable to read and having low levels of schooling is closer to that found in the control groups of those studies (literate, readers, control, after training, with one or more years of schooling), our hypothesis that previous work underestimates illiterates' NWR scores (because they were presented with an overt task, rather than a game) would be supported. Note that, alternatively, comparatively lower scores would be compatible with our other hypothesis, that there are long-term effects of low levels of infant-directed speech, particularly if lower scores are observed even among Tsimane' adults who can read.

Our final question pertains to changes with development and experience. We compare the distribution of NWR scores in children and adults using an unpaired *t*-test. Additionally, we describe within-group changes as a function of age, literacy, and formal education using unpaired t-tests and Spearman correlations.

## Results

### Comparing children's NWR scores against previous work

Fig 3 shows individual participants' NWR scores (in both word and phoneme scoring) separately for adults and children, averaged across all items. Children's NWR scores spanned from 27 to 100% for the word-level scoring, but they were a great deal less variable across participants when phoneme scoring was considered, ranging from 63 to 100%.

Collapsing across ages, children's NWR scores averaged 54.62%, meaning that children on average repeated about half of the items correctly. To increase interpretability by taking into account word complexity, we next considered only bisyllabic items. The mean for bisyllabic items in particular was 61.34%; NWR scores ranged from 16.67 to 100.00%. The low end of this is descriptively lower than previous reports, even though some of the previous reports come from longer non-words: [25] reports that individual 4-year-olds' scores on bisyllabic items ranged from 45 to 91% correct; and the 95% pseudo-confidence interval (estimated as 2 SD above and below the mean) for [22]'s 4- and 5-year-olds repeating 1- to 6-syllable long items is 40-100%.

### Comparing adults' NWR scores against previous work

Word-level scoring (averaging across all attempts and items within participants, and then averaging over participants) revealed that adults' NWR scores ranged from 36.84 to 90.91% whereas the phoneme-level scoring was even less variable than among children, ranging from 75.21 to 94.55%. The overall mean was 59.95%; that for adults who reported they did not read was 54.34%; and that for adults who reported not having completed any grades was 55.22%. These averages are descriptively lower than those reported in all previous work among non-readers or illiterate participants, with the exception of [48].

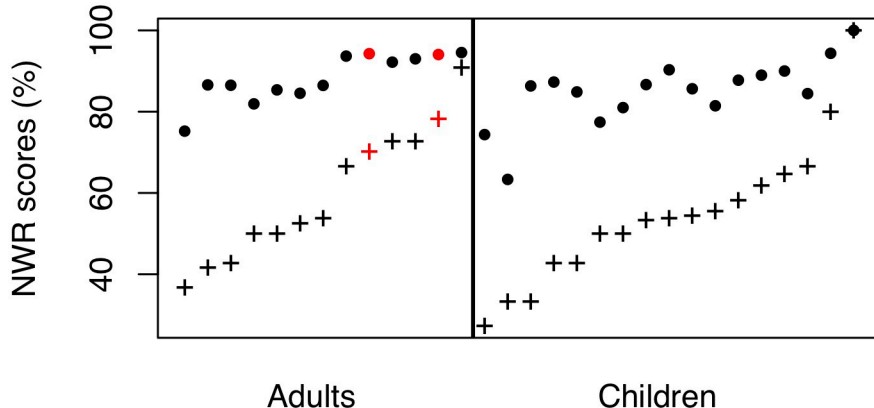

**Fig 3. NWR scores for individual participants as a function of their age group.** + indicate word-level scoring; filled circles phoneme-level scoring. Participants are sorted on the basis of their word-level score within age group. The two Tsimane' research assistants are indicated in red.

As before, to increase comparability by accounting for word complexity, we considered only bisyllabic items; as before, we average over all attempts within participants, and then over participants. Among Tsimane' adults, the mean for bisyllabic items in particular was 64.81%; NWR scores ranged from 33.33 to 100.00%. As a reminder, previous studies on illiterate, non-readers, participants having zero years of schooling, and participants tested before literacy training found about 52.86% average score overall, and 70.33% average for short words (monosyllables and bisyllables).

## Assessing the effects of age, schooling and/or literacy

As a first approach, we compared children's and adults' NWR scores with an unpaired $t$-test. This showed no significant difference between the groups, $t(26.89) = 0.85$, $p = 0.401$, with a small effect according to Cohen's $d = 0.115$. This result is not consistent with the view that NWR scores increase with age and experience beyond a certain age (for instance, via changes in phonological working memory and/or increases in vocabulary size).

In subsequent analyses, we inspect patterns of individual variability. However, results should be interpreted with some caution due to the relatively small sample size and the fact that preliminary analyses revealed no split-half reliability among children, and only weak reliability among adults (see Preprocessing section).

To assess the possibility that NWR scores are affected by literacy and/or schooling levels, we carried out additional tests. First, we separated the 7 adults who said they did not read or write from the 6 who said they did so at least a little. This analysis can only detect large effects due to low power, but a one-tailed $t$-test revealed a marginal trend for higher NWR scores in the latter, $t(11) = 1.38$, $p = 0.097$; mean for those who do not read at all = 54.34%, mean for the others = 66.50%, Cohen's $d = 0.278$. The same analysis among children revealed a smaller difference between the 10 children who did not reportedly read or write from the 5 who did: $t(13) = 0.43$, $p = 0.336$; mean for those who do not read at all = 55.39%, mean for the others = 59.68%, Cohen's $d = 0.086$. Readers are reminded that there were no children reported to read/write well, whereas 3 adults did report doing so.

Next, we inspected Spearman correlations between NWR scores and highest grade completed. An analysis among the adults suggested a non-significant trend for higher NWR scores when they had completed higher grades, r(11) = 0.448, $p = 0.125$; the estimate was smaller and

in the opposite direction among children, r(13) = -0.210, $p$ = 0.453. Readers are reminded that the range in highest grade completed was smaller among children than among adults.

## Discussion

This paper reports on exploratory analyses carried out on data collected in the context of a non-word repetition group game. We aimed to contribute to two bodies of literature, one on the relevance of early input for the development of phonological processing, and another on the role of literacy and/or schooling. Some of these analyses involve comparisons between overall NWR scores found here and that found in previous work; others rely on comparisons between subgroups of participants (e.g., self-reported readers versus non-readers). We had hoped to contribute to the study of individual variation, particularly in terms of differences due to age, schooling, and self-reported literacy. However, we discovered upon analyzing the data that split-half reliability levels were very poor for children, and relatively poor for adults. As a result, we are particularly tentative on any conclusion that is based on correlations or sub-group comparisons in what follows, and we hope these findings spark additional research.

### The long-term effects of variation in infant-directed input quantities on phonological processing

Regarding the first strand of literature, we compared the NWR scores of Tsimane' children and adults against previous developmental and adult literature. We are cautious in our inter-pretations because our study might differ from previous ones on many dimensions, and thus this is not a perfect comparison. We observe among the Tsimane' descriptively lower levels of NWR scores in both children than adults, which is compatible with the hypothesis that early input is crucial for the emergence of short-hand-like phonological representations.

Before proceeding, it is worthwhile discussing three alternative explanations for this result. First, one may wonder whether the items we used were phonologically more complex than those employed in previous work. It is infrequent for studies to report NWR scores separately as a function of item length. Nonetheless, we found some studies that did so [25, 52, 54], and combined them with studies that used non-words that were either 1-2 or 3-4 syllables in length [31, 48, 55]. We also split up NWR scores by age, literacy (among adults), and non-word length in our data. Results are represented in Fig 4, which shows that Tsimane' participants' NWR scores were lower than those of participants in previous work (matching for age, literacy, and non-word length).

Second, it is possible that our procedure led to lower NWR scores than that used in previous work. We attempted to address this in preliminary analyses (see Preprocessing section), and found that NWR scores did not correlate with group size. However, this analysis only shows that the number of people does not matter, but not whether Tsimane' children's NWR scores would have been higher had they been tested the way children were in other NWR work. We suspect the answer is no because the youngest child in each group refused to participate, and in general children were shy. We believe that if we had isolated children and tested them one by one, alone with an unknown experimenter, their NWR scores would have been even lower, because they are never alone with strangers. We return to a related point, the question of moti-vation, further below.

Third, one could argue that people diverged from the model in ways that we classified as mispronunciation but they would have classified as allophony. It is indeed the case that some languages allow a great deal of allophony. Although there is unfortunately little research on the phonetics and phonology of the Tsimane' language, our own experience in the field is that it is not the case that there is rampant allophony and great mispronunciation tolerance among the

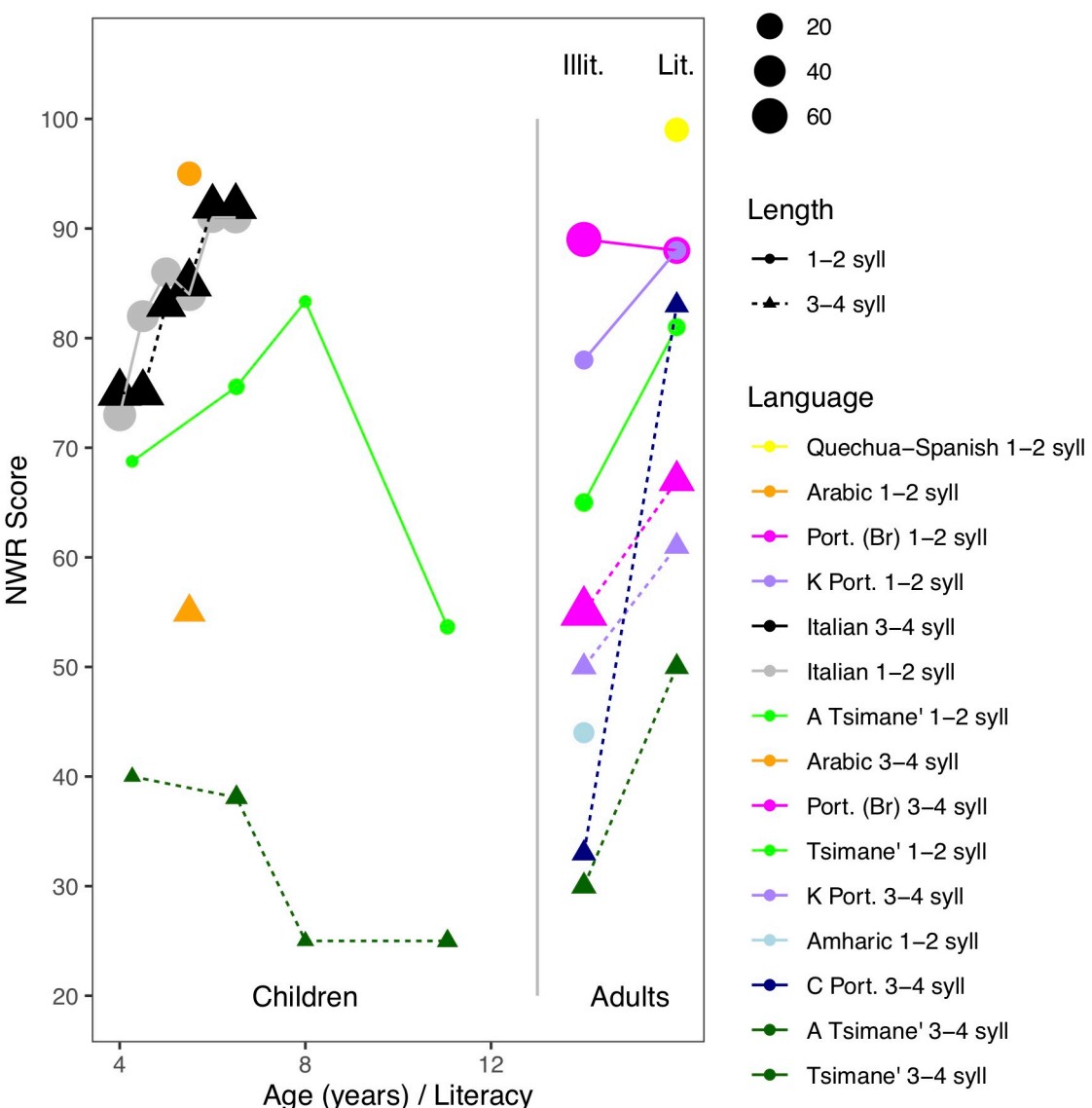

**Fig 4. NWR scores in previous and current work, splitting by participant age, non-word length, and literacy (among adults).** Study legend is sorted from highest to lowest average to facilitate linking. The size of the circle indicates sample size. Among Portuguese-speaking adults from Portugal, we separate results by study: K = [52]; C = [55].

Tsimane'. Languages allowing a great deal of allophony often have simple inventories, e.g. 3-5 vowels, but Tsimane' has a fairly complex inventory (see Fig 2), including widespread contrastive vowel nasalization, the presence of two series of central vowels, and the presence of a 4-way lingual (dental versus alveolar versus post-alveolar versus palatal) contrast. Moreover, participants sometimes attempted to repeat an item they had not repeated accurately, showing that they were aware of the mispronunciation as such.

In sum, at least based on current evidence, the data does seem to be consistent with the hypothesis that there would be long-term effects on phonological acquisition, such that children and adults with lower levels of infant-directed speech in infancy may not develop the kind of short-hand-like phonological representations and/or production schemes allowing high NWR scores. While we observe this effect here, most previous work has not found a

correlation between socio-economic status and NWR (e.g., [27, 28, 43]), even though socio-economic status is correlated with infant-directed speech quantities [12, 40]. Why this difference? Perhaps the difference in input quantity experienced by Tsimane' infants compared to that experienced by Italian [25], Swedish [27], and Slovak [22] infants is greater than the socio-economic-related input differences among Iranian [28], Swedish [27], and Israeli [43] children. Indeed, [3] finds that cultural differences in infant-directed input quantities are greater than socio-economic quantity differences within USA.

It would be interesting in future work to assess whether the prediction of lower NWR scores for children who have received low levels of infant-directed speech holds even within the Tsimane' population. To assess this, one would need to re-design our NWR test, since inspection of the stability in NWR scores in a split-half analysis revealed that children's NWR scores were unstable across the test, which may indicate that our measurement of individual variation is inaccurate. Another difficulty for this future enterprise is that the extent of stable individual variation in the amount of infant-directed speech among the Tsimane' is less known. In fact, although a few studies are coming out with direct measurements of child-directed speech estimates in more diverse populations [1, 13, 71, 72], there are very few studies that are directly comparable, even when they seem to be using similar methods. We hold hope that cross-laboratory, cross-cultural collaborations like the ACLEW Project [73] will be better able to fill this gap. It would also be interesting for future work to incorporate quality metrics (e.g., lexical diversity, [74]) and extend the study of effects from input quantity to input quality, although we are less certain of how this can be properly evaluated across very diverse languages.

## The effects of literacy and/or schooling

We also aimed to contribute data on the effects of literacy and/or schooling by comparing Tsimane' adults against previous research on non-word repetition among illiterate adults. We predicted that Tsimane' NWR scores would be higher than that observed in previous studies because we framed the exercise as an enjoyable game, rather than a scientific data collection task. However, we find descriptively lower levels of NWR scores when comparing Tsimane' adults with illiterate participants studied in previous work, even when focusing on short non-word items (mono- or bisyllables). As just discussed, overall lower levels of NWR scores in this group than others previously studied could relate to lower levels of infant-directed input for our Tsimane' adult participants when they were children (compared to non-readers and illiterates who have participated in previous work, although we know of no quantitative evaluation of infant-directed speech levels in these other populations).

To further contribute to the study of potential effects of literacy and/or schooling, we looked at individual variation among our participants. Among adults, we found a trend for higher NWR scores among those who reported they could read than those who did not; and a non-significant but moderate correlation as a function of highest grade completed. Among children, results on these relationships were less compelling, although as noted above this may be due to greater noise in the measurement at these ages. Additionally, it is possible that there is more noise than in previous studies in both our children's and adults' data because reading proficiency and schooling were self-reported rather than directly measured: Perhaps some participants reported being able to read when in fact they do not, or vice versa (see [51] for a reported correlation r = .81 between self-reported and directly measured scores in a different population).

At present, we cannot distinguish effects of literacy from potential confounds. Some previous work comparing readers and non-readers made an effort to control for various confounds, for instance drawing participants from the same families to control for e.g. genetic

predispositions and familial socio-economic status. In contrast, our present correlational design cannot control for such differences. In particular, the direction of causality may well be the opposite in our study: Adults who have developed short-hand-like phonological representations tend to learn how to read and stay longer in school than those with a different processing profile.

## Effects of input, literacy, or task oddity?

In the previous paragraphs, we have been tentative because a single observational study, in a single population, is always insufficient to prove causality: We cannot be certain of why NWR scores in this paper are lower than that found in most previous papers. Indeed, even within this paper, we point out how we cannot tease apart the effects of input and literacy, and in reality, observational comparisons across peoples can never demonstrate which feature of the people is causing a difference. Further work on many cultures, using descriptors at the level of the culture as statistical regressors may be more helpful. In this section, however, we would like to make a case for why we do not believe our data are compatible with an explanation whereby NWR scores are lower just because the task itself is difficult for all Tsimane' as opposed to because phonological processing is different than that found in other cultures studied previously with the NWR task.

A reviewer suggested that the task itself causes low NWR scores because (1) it involves an arbitrary activity, that does not have a functional role; (2) it uses non-words, which may be strange for the Tsimane'. To this we add, (3) because it is a game, rather than a structured (arbitrary) activity. We address each of these ideas, starting with the last one.

Regarding whether NWR scores would have been higher if the NWR task had been presented as a serious activity rather than a game, we start by pointing out that in the Introduction we made exactly the opposite case, that a game seemed more likely to get people relaxed than a structured activity. We know that the task was engaging for two reasons: First, several children participated several times, joining the group of their own accord even after they had already played. Second, the audio-recordings contain several examples of people laughing, not necessarily at others, but also at themselves. Their facial expressions at the time suggested that they were surprised they had trouble repeating some of the items and found that funny. In addition, we did try the more usual route, of presenting one item at a time, pre-recorded by a native speaker, and presented alongside an image, with a back story explaining that this is a word in a language that is very similar to Tsimane'. It only took a couple of tries to realize that this was extremely confusing, as we could not get a single repetition in this manner. Thus, we are convinced that the game format was preferable.

This leads us to the argument that the reason why NWR scores were relatively low is because the task was not functional. Let us start by pointing out that this is always the case. When other researchers have presented the task as "learning Martian words", this is also not functional. One way in which we could have made the task functional is if we had paid people as a function of their NWR scores or incorporated some other form of extrinsic motivation. We will return to this in the next subsection, when suggesting avenues for future research.

We thus turn to the idea that the reason why NWR scores were low is because it used non-words, which may be strange for the Tsimane' (as suggested by a reviewer). One may think that societies in which all individuals engage in many years of formal education have a larger stock of words, and in fact individual speakers do not know all the words of the language, whereas when formal schooling is more variable all speakers know all words of the language. We think this is unlikely, because in current human cultures there is always some degree of specialization, leading some people to know some specialist words that other speakers of the

same language ignore (names of plants, tools, mythical stories with specific place and people names, names for feminine versus masculine activities, etc.) It is also untrue of the Tsimane', who are in contact with mainstream Bolivian culture and other ethnic groups, and have incorporated words into their language. Loanwords must have been non-words initially, but they have been integrated into the Tsimane' language. For instance, the following are two among the several words we recognize as coming from Spanish in [62]'s dictionary: "tisira" (scissors), "castigo" (punishment). In [65]'s transcriptions of conversations, stories, and other materials, we find many more examples, notably the names of the months such as "abril" (April) and "agosto" (August), which we know to be integrated because they also appear with Tsimane' suffixes: "abrilk$^h$an" and "agostok$^h$an".

Still, some readers may not be convinced by these arguments, believing that this NWR game may not reflect people's phonological processing but instead comfort with the game. There is sufficient evidence in our data to show very clearly that NWR scores in this task *do* reflect phonological processing: NWR scores were significantly worse for longer items, and significantly worse for items containing closed syllables compared to items that contained only open syllables (see Preprocessing section). There is also evidence against the second part of that argument, since experience with the game itself did not help: We found no changes as a function of trial order or whether the item had been presented in the past (see Preprocessing section).

Finally, we can also wonder what a more appropriate task to measure phonological processing could be. Alternative choices are (1) syllable or phoneme recognition (does "pacman" contain "man" or "m"?); (2) phoneme substitution or syllable blending (what happens when you put the beginning of "tan" with the end of "pick"? possible answers being "tick" or "tack"); and (3) phonological fluency (name words starting with the sound "p"). Except for syllable recognition, all of these rely on phonemes as basic units, and there is discussion as to whether phonemes are consciously accessible units for preliterate children and illiterate adults (and even literate adults who use an ideographic script; e.g. [75]). Also, all of these tasks are metalinguistic, requiring the participant to understand what "sound" is and what "contain" is in the context of words. Although it would require a systematic review to establish this with certainty, our impression is that non-word repetition is much more common in the literature on preliterate children's phonological processing than any of these other tasks except in the context of learning to read or school readiness.

All of that said, we do not want to leave the reader with the impression that we have proved causality. Lower scores in one task in one population could be due to a host of reasons. For example, in a recent paper on perceptual skills among school-aged children aboriginal to Australia, it was found that children scored relatively poorly in phonological awareness, and individual variation in these scores correlated with individual variation in psychoacoustic tests [76]. We have not tested hearing among the Tsimane', and it is possible that a higher proportion of Tsimane' have hearing difficulties than other populations represented in the literature, and that variation in auditory skills would impact their NWR scores. This is then a potential confound that would need to be addressed in future work.

## Additional considerations for future research

Interestingly, we did not find significant differences as a function of age, among children or even in a comparison between children and adults. Although as just mentioned the correlation within childhood is compromised by the low reliability of the measure, the lack of significant increase between childhood and adulthood is more surprising. Observing this, we revisited the developmental literature summarized in Figs 1 and 4, which shows age-related changes in the

order of 10-20 percentage points among children (Fig 1), which is certainly larger than the 5% found here between children and adults. However, notice that for many of the children samples represented in Fig 1, the age range includes the onset of schooling/literacy training, and thus the 10-20% increase could be due to age and/or schooling/literacy. In fact, Fig 4 shows wide variability in NRW scores among adults. For instance, Portuguese-speaking literate adults presented with 3- or 4-syllable non-words scored less than 70% in one study [54] and over 80% in another [55]. It is unfortunate that no previous study included groups of adult and child participants tested on the exact same materials, and thus differences could still be due to complexity of the non-words used (e.g., presence of consonant clusters or codas). We therefore encourage other researchers to systematically collect data from children and adults drawn from the same population, to have a more precise idea of what children's and adults' scores are when the exact same stimuli are used.

Before closing this paper, we wanted to return to one of its key contributions, namely the idea of incorporating a measure of phonological processing in the form of a group game. The game was neither cooperative nor competitive, and we can imagine advantages and disadvantages to creating versions that pit individuals or groups against each other. In particular, it is worthwhile reflecting on the fact that when children or adults come into our laboratories, they may be motivated to cooperate with us and provide high NWR scores. In child studies, each repetition (regardless of whether it is correct or not) may be rewarded with an interesting change in a visual display or a sticker on a card. Equivalently, it is possible that previous studies on literate and illiterate adults drew from a population of participants who were intrinsically motivated to aid science or to contribute to the discovery of diagnostics for cognitive aging. In contrast, we did not provide our participants with any extrinsic motivation, and we did not check for any specific intrinsic motivation. It may be relevant to discuss this with the participants in the future to increase comparability in the level of motivation, particularly in cross-cultural studies.

## Conclusions

To conclude, we collected non-word repetition data from Tsimane' children and adults to contribute to two lines of research. First, we found descriptively lower levels of NWR scores in both children and adults compared to previous work, which is consistent with the hypothesis that there are long-term effects on phonological processing of experiencing low levels of directed input in infancy. Second, we found some evidence that literacy and/or schooling increases NWR scores in non-word repetition, although these results should be interpreted with caution given the small sample size. Finally, we did not find strong age effects within childhood or in a comparison of children and adults. Altough we believe these data are important to share because of the rarity with which populations like the Tsimane' are allowed participation in psycholinguistic research, we also pointed out limitations both in terms of the data collected and the comparisons against previous work.

## Acknowledgments

We are grateful to Radhia Achheb, Michel Dutat, Vireack Ul, and Catherine Urban for logistical assistance in organizing the field trip; to Angèle Barbedette and Xuan Nga Cao for setting up analysis pipelines and constructing parsable versions of Tsimane' written materials which were crucial in the design of the items; to the three Tsimane' research assistants who helped us prepare and deliver the stimuli; and to all the Tsimane' individuals who participated in the study.

## Author Contributions

**Conceptualization:** Alejandrina Cristia, Gianmatteo Farabolini.

**Data curation:** Alejandrina Cristia, Gianmatteo Farabolini, Camila Scaff.

**Funding acquisition:** Alejandrina Cristia, Jonathan Stieglitz.

**Investigation:** Alejandrina Cristia, Jonathan Stieglitz.

**Methodology:** Alejandrina Cristia.

**Project administration:** Alejandrina Cristia, Camila Scaff, Jonathan Stieglitz.

**Resources:** Alejandrina Cristia, Gianmatteo Farabolini, Camila Scaff, Jonathan Stieglitz.

**Software:** Alejandrina Cristia.

**Supervision:** Naomi Havron.

**Validation:** Alejandrina Cristia.

**Visualization:** Alejandrina Cristia, Naomi Havron.

**Writing – original draft:** Alejandrina Cristia, Gianmatteo Farabolini, Naomi Havron.

**Writing – review & editing:** Alejandrina Cristia, Gianmatteo Farabolini, Camila Scaff, Naomi Havron, Jonathan Stieglitz.

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
