## [Decision Letter · Decision Letter 0]

24 Feb 2020

PONE-D-19-31695

Infant-directed input and literacy effects on phonology: Non-word repetition accuracy among the Tsimane'

PLOS ONE

Dear Dr. Cristia,

Thank you for submitting your manuscript to PLOS ONE. After careful consideration, we feel that it has merit but does not fully meet PLOS ONE’s publication criteria as it currently stands. Therefore, we invite you to submit a revised version of the manuscript that addresses the points raised during the review process.

Both reviewers note that the paper has value and appeal to the field, particularly in investigating a non-WEIRD population that is extremely underrepresented in the literature. At the same time, they highlight several significant issues with the framing of the paper, breadth of literature covered, and interpretations drawn. I agree that these issues should be addressed to make the submission suitable for publication. I have summarized the main points made by each reviewer, with their specific comments below.

Both reviewers agree that there are gaps in the literature covered and recommend incorporating additional literature, as well as expanding upon some of the studies that are already mentioned. Specifically, Reviewer 1 asks for more coverage of the research relating to literacy and the development of phonological knowledge, and discussion of the quality (rather than just quantity) of children's language input on language development. Reviewer 2 asks for more information about the Tsimane language (particularly regarding the phonological inventory and allophonic variation), amount of child language input across different communities, results from additional nonword repetition tasks, and discussion of other research that has tested phonological awareness in remote communities (e.g., Sharma, Wigglesworth, Savage & Demuth, 2020).

While you do explicitly acknowledge that your analyses are exploratory, Reviewer 1 advises that this should be addressed from the outset of the paper. Reviewer 2 expresses concern that the interpretations are still overextended given that it is not yet established whether the NWR task here reliably measures phonological skills, and given the confounds in the task and population within the experiment and relative to prior literature. This is particularly with regard to the question of amount of early language input.

Still, as Reviewer 2 points out, this paper reflects the challenge of adapting typically lab-based methods to the field in a remote setting, and in that has significant value to others looking to test in underrepresented populations. As such, Reviewer 2 recommends lending more discussion to the process/challenges that were certainly faced in such an endeavor to help future researcher anticipate and accommodate similar challenges.

We would appreciate receiving your revised manuscript by Apr 09 2020 11:59PM. To enhance the reproducibility of your results, we recommend that if applicable you deposit your laboratory protocols in protocols.io, where a protocol can be assigned its own identifier (DOI) such that it can be cited independently in the future. For instructions see: http://journals.plos.org/plosone/s/submission-guidelines#loc-laboratory-protocols

We look forward to receiving your revised manuscript.

Kind regards,

Karen E. Mulak, Ph.D.

Academic Editor

PLOS ONE

Journal Requirements:

2. Please provide additional details regarding participant consent. In the Methods section, please state how verbal consent was recorded and whether the ethics committee approved this consent procedure. We note that consent was obtained from the parents of the minors that took part in the study. Please clarify whether provided explicit permission for their children to take part in research.

3. Please note that PLOS ONE uses a single-blind peer review procedure. We would therefore be grateful if you could include in the information that has been anonymised for peer review in the manuscript.

Reviewers' comments:

Reviewer's Responses to Questions

**Comments to the Author**

1. Is the manuscript technically sound, and do the data support the conclusions?

Reviewer #1: Partly

Reviewer #2: Partly

2. Has the statistical analysis been performed appropriately and rigorously? 

Reviewer #1: Yes

Reviewer #2: I Don't Know

3. Have the authors made all data underlying the findings in their manuscript fully available?

Reviewer #1: Yes

Reviewer #2: Yes

4. Is the manuscript presented in an intelligible fashion and written in standard English?

Reviewer #1: Yes

Reviewer #2: Yes

5. Review Comments to the Author

Reviewer #1: The basic idea of the paper is fantastic, and is important, so I would definitively like it to be published. However, it suffers from several limitations.

The first and more important one is that it is exploratory, and based on very small samples. Yet, except for a rapid note in the method (line 231, p. 6, “All analyses must thus be considered as exploratory), we must wait until line 527, in the discussion, to see that the Authors acknowledge that “This paper reports on exploratory analyses ….”. This should be acknowledged much earlier in the paper, already in the Introduction. The idea the Authors explore and the challenge this represents makes it perfectly acceptable to present an exploratory paper. Also, it is surprising that reading ability of the participants was not tested, but self-reported. This limitation should also be clearly acknowledged.

Second, there are some important details and references missing as regards the effects of literacy on nonword repetition. For instance, in discussing the effects of literacy vs. schooling on memory, the authors should mention that there is at least one longitudinal study reporting a small but significant improvement in phonological memory (nonword repetition, with NO mixing with real words) on seven Portuguese adults who were unschooled and fully illiterate at the beginning of the study, but were successfully taught to decode in 3 months (Kolinsky et al. 2018, Reading & Writing). This improvement in nonword repetition was correlated positively with progress in phoneme awareness, as assessed by a phoneme deletion task.

Also note that performance on nonword repetition at the beginning of study (i.e., when participants were still fully illiterate, as it was the case in the two pre-tests, T1 and T2) was very low. Interestingly, though, their performance seems higher than the one observed in the present study, at least when items of similar length are considered:65,62% on average on items with 1 to 4 syllables, 62,5% on average on disyllabic items.

As regards the effect of item length, line 548 p. 13 it is said that “It is infrequent for studies to report performance separately as a function of item length”. We agree with that, and this is why it would be important to compare the data of the present study with those observed by Kolinsky et al. (2018) in fully illiterate Portuguese adults (see their Appendix): as I already reported, at pretest their participants presented 65,62% on average on items with 1 to 4 syllables, 62,5% on average on disyllabic items.

Also, presentation of the data is unsatisfactory as regards the effect of item length. Indeed, average group values for each nonword length are not presented (only individual scores in Figure 4), and, no separate scores are provided for the “readers” vs. “non-readers”. This would be very interesting, as there is no reason to expect a huge (readers vs non-readders) group difference on short (mono or disyllabic) items, but such a group difference is expected on longer items (perhaps already 3-syllables long; probably 4-syllabes long items).

In any case, in the abstract, the formulation “we found weak evidence that literacy and/or education improves performance in non-word repetition” is ambiguous; please reformulate. On lines 648-650 (p. 15), it is much more exactly said that “(..), we found some evidence that literacy and/or education improves performance in non-word repetition, although these results should be interpreted with caution given the small sample size”.

Finally, the authors should not restrict the discussion to the quantity of verbal input provided to children, but include the quality of that input (see discussion in Golinkoff et al., Child Development 2019, as regards quality speech directed to children rather than overheard speech).

Minors and typos

Table 1: presentation is not clear, in particular as whether schooling was controlled for (e.g., all the “literate” participants in Castro-Caldas et al 1998 had attended school) . More generally, the terminology adopted in that table as well as on p. 4 (bottom) is confusing: the Authors speak about "illiterate", "non-readers" and "non-literates". What is the difference ?(I suspect none).

Line 61 p. 3: correct “enviroments” : “environments”

“Castro” should be replaced by “Castro-Caldas” in table 1 (=his last name)

Replace “Louriero” by “Loureiro” in table 1

Loureiro et al 2004 is missing in the Reference list

In table 1 the performance for Castro-Caldas 1998 is not for long nonwords, but average performance

Line 424 on p. 10: “paraticipants” should be replaced by “participants”

Line 519, p. 12 “aduls” should be changed into “adults”

Reviewer #2: This study about language experience in the Tsimane community and its role in phonological representations. While it is an interesting project, and important in terms of addressing child language in a community which offers a different perspective, there is certainly work required on the interpretation in this study. I would not want to see this published without some rethinking on what the results could really mean. It is not currently clear if the findings are truly about long-term effects on phonological representations, or simply task effects i.e. the actual experiment - framed as a game - is not a typical kind of activity for this community. That is OK, but the findings are currently a little opaque. It is not currently clear if the experiment is useful for highlighting people's real abilities. Also, some languages allow a lot of variation in phonetic realisations, so how can we be certain that "errors" are truly "errors" and whether the results are truly shining a light on phonological representation? I feel more evidence or justification is required.

I list my specific comments below:

In the introductory sections it would be very useful to understand the specifics of the Tsimane language - where exactly is it spoken (a map would be very helpful), how many speakers, what language family, and what is the phonological inventory (especially important given the topic). Also just a mention of the different spellings of the language name would be useful.

p.2 line 60 onward - here I was expecting a nod to research which has already addressed this topic, such as work by Katherine Demuth.

p.2 line 37 - technically, children are exposed to language before birth, in utero. I take your point that children are not "pre-programmed" in general, but I would reword this a little.

p.3 line 55 - the non-word "beng" should have italics or stand out from the main text, it looks a little like a typo currently.

p.3 line 97 - I am not sure why "anonymized" is not just listed the author's name? The document I have is not anonymized otherwise.

p.3 line 99 onward. I think the approach of doing a google scholar search in this way to understand past literature is not fully appropriate. I think a proper / comprehensive literature review, not excluding any papers because they were not in the top 20 "hits", would be more useful.

p.4 I am not convinced that the word "performance" should be used to talk about people's ability in their own languages. It seems that the language is used for day-to-day communication, so maybe "performance" is not the right word for this.

p.4 line 158 - I think it is a really crucial point that "arbitrary tasks" are not typical for people with little education. This really needs to be explored further.

p.6 line 210 I take the point about the critical period, but I believe it is also true that children can sound native in another language up until around 8 - so perhaps a bit more discussion / more references here would be useful.

p. 6 Methods - I think the work around the methodology needs more discussion. It sounds like actually a lot of work was done in difficult circumstances, and that various changes had to be made int he course of executing the study. This would be useful for other researchers to know about.

p.6 line 235, p.7 line 240, line 243 - again "Anonymized" is not really needed here.

p.7 Here is where it would be useful to reflect back on the phonological inventory of the language. It would also be especially useful to know, in general, how frequent the sound patterns are - is the palatal especially common, how much phonetic variation is allowed, and was this incorporated into the design of the project? Things like this help to better understand the context.

p.8 line 312 - reference 49 is incomplete

p.9 In the "scoring" section, any time a non-word is listed it should be in italics, or separated from the main text somehow for readability.

p.9 line 375 - I would consider replacing "refusing to make an attempt" with "not making an attempt".

p.9 line 376 - presumably the issue about not being able to determine a speaker's identity in your recordings is a good learning opportunity that could be described / explored a little further in the paper. What solutions could you offer other researchers who have to use a similar experimental condition? Perhaps individual lapel microphones? Obviously it is very important to be sure which data belongs to which person.

p.10 in terms of what affects "performance", I think we could consider phonological complexity and also frequency of certain sound patterns as potential factors. I think this should be addressed in some way.

p.12 line 495 - what is similar about your study and [39]? This would be interesting to know and could shed light on what your results mean.

p.13 line 540 I am not sure it is possible to compare this work with past developmental literature. Your work could be considered in light of what is out there, and can offer some contributions, but the ways the studies are conducted (necessarily very differently) means they are not in any way directly comparable.

p.13 line 544-553 It is not really possibly to compare this work from work in literate cultures. At one point in this study there is a discussion about how the items in the Tsimane study may be harder than those used in previous work, but I don't think currently that the study can officially say these items are on par with previous work at all. The items probably are indeed harder conceptually for Tsimane people, as they are non-words, and non-words are strange for people, especially in a place where education is not common (and presumably language play is also not a known concept?).

p.13 line 560 - as it stands, it sounds a bit unethical to approach very young children who "systematically refused to participate". Was it simply that they didn't want to? Maybe this needs to be reworded.

p.14 line 586 - there is so much research about the amount of talk children are exposed to, in different communities worldwide (using, e.g. Lena) - it is definitely important to review some of that here for more context.

p.14 line 591 onward - if the society is overall not very literate, then it seems almost impossible to separate the task effect (i.e. it is a strange task for them to participate in) from the "low" performance.

p.14 line 610 - what did reference 36 find? More detail needed here.

p.14 line 615 - there has not been any real discussion of potential individual differences in the paper - I feel it would be useful here.

p.14 line 619 - if there are little differences between children and adults, then how do we really know that the experiment is getting accurate results? It is hard to be convinced about the task at this stage, because it really does seem like there is a "low reliability of the measure". If there really is clear evidence that the task works (such as the differences between lower vs better educated Tsimane people), then readers will need some more convincing about this point.

6. PLOS authors have the option to publish the peer review history of their article (what does this mean?). If published, this will include your full peer review and any attached files.

Reviewer #1: No

Reviewer #2: Yes: Dr. Debbie Loakes

---

## [Author Response · Author response to Decision Letter 0]

16 May 2020

Please see response to reviewers for detailed point by point reply

---

## [Decision Letter · Decision Letter 1]

22 Jun 2020

PONE-D-19-31695R1

Infant-directed input and literacy effects on phonological processing: Non-word repetition scores among the Tsimane'

PLOS ONE

Dear Dr. Cristia,

Thank you for submitting your manuscript to PLOS ONE. After careful consideration, we feel that it has merit but does not fully meet PLOS ONE’s publication criteria as it currently stands. Therefore, we invite you to submit a revised version of the manuscript that addresses the points raised during the review process.

I agree with Reviewer 1 and your own assessment that the paper is very much improved this round. There still remain a few minor points to be addressed. Reviewer 1 has pointed out a discrepancy in the reported averages for adults between Table 1 and later on in the text which will need to be corrected, as well as two additional minor points. Reviewer 1 also suggests moving Figs 3-6 to supplementary material, and I echo this suggestion, particularly with regards to Figs 3 and 4, leaving Figs 5 and 6 to your discretion. Unfortunately Reviewer 2 was unable to review the paper a second time around. However, I have reviewed the paper myself and concluded that you have adequately addressed the concerns brought up in the last round. I also add a few minor points:

-p. 2;24: "representations that makes" <-- "representations that make"

-Please standardize spelling of "Tsimane". It variably appears with and without a final apostrophe (unless there is a distinction I am missing).

-Please italicize statistical variables (e.g., *t, p*).

-p. 15;612: The mean for those who do not read at all should go to two decimal places.

-p. 16;671: I found the phrasing around the speculation as to whether "mispronunciations are tolerated" in Tsimane a bit odd. It sounds as if the language allows mispronunciations of words, which of course is contradictory since they would then not be mispronunciations. I would recommend rephrasing it a bit to specify that *what you classified as a mispronunciation in your task* may reflect acceptable allophonic variation in the Tsimane language.

We look forward to receiving your revised manuscript.

Kind regards,

Karen E. Mulak, Ph.D.

Academic Editor

PLOS ONE

Reviewers' comments:

Reviewer's Responses to Questions

**Comments to the Author**

1. If the authors have adequately addressed your comments raised in a previous round of review and you feel that this manuscript is now acceptable for publication, you may indicate that here to bypass the “Comments to the Author” section, enter your conflict of interest statement in the “Confidential to Editor” section, and submit your "Accept" recommendation.

Reviewer #1: (No Response)

2. Is the manuscript technically sound, and do the data support the conclusions?

Reviewer #1: Yes

3. Has the statistical analysis been performed appropriately and rigorously? 

Reviewer #1: N/A

4. Have the authors made all data underlying the findings in their manuscript fully available?

Reviewer #1: Yes

5. Is the manuscript presented in an intelligible fashion and written in standard English?

Reviewer #1: Yes

6. Review Comments to the Author

Reviewer #1: Congratulations for the great job the authors did !

Yet there is a bug in the data, at least the adult ones

In Table 1 it is said that average values are 53% (n = 7) vs. 71% (n = 6) for adult nonreaders vs. readers

But p. 15, lines 611-612 , it is said that mean for those who do not read at all = 66.5%, mean for the others = 54.34.

First, this indicates HIGHER scores in nonreaders.

Second, it is not as simple inversion of the Table 1 data, so please fix this problem.

Minor points:

Legend of table 1: should indicate that NWR scores are %

Fig 2 should rather be called a table

More generally, the Figures are too numerous.

I would suggest presenting figures 3, 4, 5, 6 as supplemental material, not in the main text

This, together with the conversion of Fig 2 into a table, would leave a total of 3 figures (FIG1 FIG 7 FIG 8)

7. PLOS authors have the option to publish the peer review history of their article (what does this mean?). If published, this will include your full peer review and any attached files.

Reviewer #1: No

---

## [Author Response · Author response to Decision Letter 1]

3 Jul 2020

Please see cover letter with color coding and formatting. Below, our replies are preceded by >>

Dear Dr. Mulak,

Thank you for the fast turn-around time and for stepping in as a reviewer. We know how difficult it is to find time for service, and we truly appreciate your editorial work.

We have revised the manuscript to address all suggestions, which in summary are:

Move 3 figures and the corresponding analyses to supplementary materials

Corrected typos and rephrased an ambiguous comment.

Thoroughly checked all formulas in the manuscript. In doing so, we changed analyses to always first average within individuals, and only then across individuals (before some of the averages were done directly over all data, effectively leading individuals with more data to have a greater weight, thus leading to differences in results across analyses).

We have not renamed Fig 2 as a Table, because converting it into a table would have meant a risk of these symbols being converted in some displays, whereas this way we know exactly what readers will see. 

We look forward to hearing from you,

The authors

PONE-D-19-31695R1

Infant-directed input and literacy effects on phonological processing: Non-word repetition scores among the Tsimane'

PLOS ONE

Dear Dr. Cristia,

Thank you for submitting your manuscript to PLOS ONE. After careful consideration, we feel that it has merit but does not fully meet PLOS ONE’s publication criteria as it currently stands. Therefore, we invite you to submit a revised version of the manuscript that addresses the points raised during the review process.

I agree with Reviewer 1 and your own assessment that the paper is very much improved this round. There still remain a few minor points to be addressed. Reviewer 1 has pointed out a discrepancy in the reported averages for adults between Table 1 and later on in the text which will need to be corrected, as well as two additional minor points. Reviewer 1 also suggests moving Figs 3-6 to supplementary material, and I echo this suggestion, particularly with regards to Figs 3 and 4, leaving Figs 5 and 6 to your discretion. 

>> We have incorporated all suggestions by Reviewer 1. Please see detailed point-by-point replies below.

Unfortunately Reviewer 2 was unable to review the paper a second time around. However, I have reviewed the paper myself and concluded that you have adequately addressed the concerns brought up in the last round. 

>> Thank you so much for that! We truly appreciate it.

I also add a few minor points:

-p. 2;24: "representations that makes" <-- "representations that make"

>> Fixed, as were all other typos in a thorough read.

-Please standardize spelling of "Tsimane". It variably appears with and without a final apostrophe (unless there is a distinction I am missing).

>> Fixed, as were all other typos in a thorough read. We should note that we have not changed the spelling of Tsimane' in the references, because that would entail changing the title of previous papers.

-Please italicize statistical variables (e.g., t, p).

>> Fixed, as were all other typos in a thorough read.

-p. 15;612: The mean for those who do not read at all should go to two decimal places.

Fixed, as were all other typos in a thorough read.

-p. 16;671: I found the phrasing around the speculation as to whether "mispronunciations are tolerated" in Tsimane a bit odd. It sounds as if the language allows mispronunciations of words, which of course is contradictory since they would then not be mispronunciations. I would recommend rephrasing it a bit to specify that what you classified as a mispronunciation in your task may reflect acceptable allophonic variation in the Tsimane language.

>> Thank you for the recommended rewriting, we have incorporated it. The sentence now reads: "one could argue that people diverged from the model in ways that we classified as mispronunciation but they would have classified as allophony."

We look forward to receiving your revised manuscript.

Kind regards,

Karen E. Mulak, Ph.D.

Academic Editor

PLOS ONE

Reviewers' comments:

Reviewer's Responses to Questions

Comments to the Author

1. If the authors have adequately addressed your comments raised in a previous round of review and you feel that this manuscript is now acceptable for publication, you may indicate that here to bypass the “Comments to the Author” section, enter your conflict of interest statement in the “Confidential to Editor” section, and submit your "Accept" recommendation.

Reviewer #1: (No Response)

2. Is the manuscript technically sound, and do the data support the conclusions?

Reviewer #1: Yes

3. Has the statistical analysis been performed appropriately and rigorously?

Reviewer #1: N/A

4. Have the authors made all data underlying the findings in their manuscript fully available?

Reviewer #1: Yes

5. Is the manuscript presented in an intelligible fashion and written in standard English?

Reviewer #1: Yes

6. Review Comments to the Author

Reviewer #1: Congratulations for the great job the authors did !

>> Thank you!

Yet there is a bug in the data, at least the adult ones

In Table 1 it is said that average values are 53% (n = 7) vs. 71% (n = 6) for adult nonreaders vs. readers

But p. 15, lines 611-612 , it is said that mean for those who do not read at all = 66.5%, mean for the others = 54.34.

First, this indicates HIGHER scores in nonreaders.

Second, it is not as simple inversion of the Table 1 data, so please fix this problem.

>> We apologize for this error! It was indeed an inversion, complemented with the fact that the table was an average over all items (without first averaging within participants) and the second was an average over participants first then across participants. We have now thoroughly checked the code to make sure this and any other issues are resolved. 

Minor points:

Legend of table 1: should indicate that NWR scores are %

>> We have added this.

Fig 2 should rather be called a table

>> We have kept this figure as such, because turning it into a table could have caused typographical errors. This way we have full control over the symbols as they are displayed.

More generally, the Figures are too numerous.

I would suggest presenting figures 3, 4, 5, 6 as supplemental material, not in the main text

>> We have moved these figures to the supplementary material document.

This, together with the conversion of Fig 2 into a table, would leave a total of 3 figures (FIG1 FIG 7 FIG 8)

7. PLOS authors have the option to publish the peer review history of their article (what does this mean?). If published, this will include your full peer review and any attached files.

Do you want your identity to be public for this peer review? For information about this choice, including consent withdrawal, please see our Privacy Policy.

Reviewer #1: No

---

## [Editor Report · Decision Letter 2]

15 Jul 2020

PONE-D-19-31695R2

Infant-directed input and literacy effects on phonological processing: Non-word repetition scores among the Tsimane'

PLOS ONE

Dear Dr. Cristia,

Thank you for submitting your revision to PLOS ONE. Apologies if there was some confusion with the comment regarding the preprocessing analyses in the last version. The text corresponding to the analyses can stay in the document; it is only the figures that were suggested to be moved to supporting information. Following the PLOS ONE guidelines for supporting information, could you include the supporting figures in your submission and include the supporting figure captions at the end of your manuscript.

In addition, on page 11, line 453, the value "2" was changed to "0.00%." I am wondering if this was a typo. I also noticed in passing that on page 14, line 618, Cohen's *d* should also be italicized.

Once I receive these minor revisions I anticipate the manuscript will be suitable for publication.

We look forward to receiving your revised manuscript.

Kind regards,

Karen E. Mulak, Ph.D.

Academic Editor

PLOS ONE

---

## [Author Response · Author response to Decision Letter 2]

29 Jul 2020

Dear Dr. Mulak,

Thank you once more for the fast turn-around time and your careful editorial work. 

We have revised the manuscript to address all suggestions, which are:

Cohen's d is Italicized

Corrected a typo (2% → 0%).

Moved back the text from the online supplementary document into the Preprocessing section.

We have not listed supplementary figures, because we are referring to an external resource, which is a pdf document containing the figures as well as other text and other analyses. We read carefully the section of instructions referring to supplemental materials, and while we appreciate the offer for Plos to store these materials, they are already archived in a scientific repository, together with all the rest of the supplementary materials (data, scripts). We fear that repeating this document in two places may cause confusion. 

We look forward to hearing from you,

The authors

##########################################

PONE-D-19-31695R2

Infant-directed input and literacy effects on phonological processing: Non-word repetition scores among the Tsimane'

PLOS ONE

Dear Dr. Cristia,

Thank you for submitting your revision to PLOS ONE. Apologies if there was some confusion with the comment regarding the preprocessing analyses in the last version. The text corresponding to the analyses can stay in the document; it is only the figures that were suggested to be moved to supporting information. 

> We have moved the text back into the paper and corrected all references to these analyses, as well as clarified that we are not referring to supplementary figures but rather an online supplementary resource.

Following the PLOS ONE guidelines for supporting information, could you include the supporting figures in your submission and include the supporting figure captions at the end of your manuscript.

> We have not listed supplementary figures, because we are referring to an external resource, which is a pdf document containing the figures as well as other text and other analyses. We read carefully the section of instructions referring to supplemental materials, and while we appreciate the offer for Plos to store these materials, they are already archived in a scientific repository, together with all the rest of the supplementary materials (data, scripts). We fear that repeating this document in two places may cause confusion.

In addition, on page 11, line 453, the value "2" was changed to "0.00%." I am wondering if this was a typo. 

> Sorry for that! It was indeed a typo, now corrected.

I also noticed in passing that on page 14, line 618, Cohen's d should also be italicized.

> We fixed that and all other d's.

Once I receive these minor revisions I anticipate the manuscript will be suitable for publication.

> This is great news, thank you!

 > We are submitting all three items

> We don't need to revise this

> This has already been done in the previous round of reviews.

> We have uploaded data, scripts, and further materials onto the scientific repository Open Science Framework.

We look forward to receiving your revised manuscript.

Kind regards,

Karen E. Mulak, Ph.D.

Academic Editor

PLOS ONE

While revising your submission, please upload your figure files to the Preflight Analysis and Conversion Engine (PACE) digital diagnostic tool, https://pacev2.apexcovantage.com/. 

> This has already been done in the previous round of reviews.

PACE helps ensure that figures meet PLOS requirements. To use PACE, you must first register as a user. Registration is free. Then, login and navigate to the UPLOAD tab, where you will find detailed instructions on how to use the tool. If you encounter any issues or have any questions when using PACE, please email PLOS at figures@plos.org. Please note that Supporting Information files do not need this step.

---

## [Editor Report · Decision Letter 3]

3 Aug 2020

Infant-directed input and literacy effects on phonological processing: Non-word repetition scores among the Tsimane'

PONE-D-19-31695R3

Dear Dr. Cristia,

We’re pleased to inform you that your manuscript has been judged scientifically suitable for publication and will be formally accepted for publication once it meets all outstanding technical requirements.

Kind regards,

Karen E. Mulak, Ph.D.

Academic Editor

PLOS ONE
---

## [Editor Report · Acceptance letter]

28 Aug 2020

PONE-D-19-31695R3 

Infant-directed input and literacy effects on phonological processing: Non-word repetition scores among the Tsimane' 

Dear Dr. Cristia:

I'm pleased to inform you that your manuscript has been deemed suitable for publication in PLOS ONE. Congratulations! Your manuscript is now with our production department. 

Kind regards, 

on behalf of

Dr. Karen E. Mulak 

Academic Editor

PLOS ONE